# An Analysis of Personal Learning Environments and Age-Related Psychosocial Factors of Unaccompanied Foreign Minors

**DOI:** 10.3390/ijerph17103700

**Published:** 2020-05-24

**Authors:** María Tomé-Fernández, Marina García-Garnica, Asunción Martínez-Martínez, Eva María Olmedo-Moreno

**Affiliations:** Department Methods of Research, Faculty of Education, University of Granada, 10871 Granada, Spain; mariatf@ugr.es (M.T.-F.); asuncionmm@ugr.es (A.M.-M.); emolmedo@ugr.es (E.M.O.-M.)

**Keywords:** unaccompanied foreign minors, personal learning environment, psychosocial factors, social networking

## Abstract

Spain is one of the countries with the greatest influx of immigrants and, specifically, of unaccompanied foreign minors (UFMs). The educational and social inclusion of unaccompanied foreign minors poses both a challenge and a threat to current policy. Nonetheless, studies linking educational aspects to the phenomenon of the integration of these children are scarce and do not specify the most influential educational tools and strategies. In this sense, a descriptive, quantitative and cross-sectional research study is presented. The aim of this study is to examine whether variables such as age and the use of applications and social networks determine the personal learning environments (PLE) of unaccompanied foreign minors. The sample of the present study was formed by 624 individuals (♂ = 92.1% (*n* = 575); ♀ = 7.9% (*n* = 49)) aged between 8 and 17 years old. The majority came from Morocco and resided in the cities of Ceuta and Melilla. The “PLE and Social Integration of UFMs” questionnaire was used as the study instrument. Amongst the main findings, significant differences are highlighted in the personal learning environments as a function of age-related psychosocial factors as they pertain to unaccompanied foreign minors. Four factors were seen to exist in relation to the personal learning environments of unaccompanied foreign minors: self-concept of the learning process, planning and management of learning, use of resources and tools, and communication and social interaction. The same trend was observed in the four factors, with older age groups reporting better scores. On the other hand, results show that the use of applications and social networks have a significant and favourable impact on personal learning environment construction.

## 1. Introduction

By a long way, Spain is one of the European countries with the greatest influx of immigrants, above all those coming from Latin America, Asia, Sub-Saharan Africa and Maghreb [1,2]. Specifically, the majority of immigrants arriving in Spanish cross-border cities [3] come from this latter location [4,5].

Since the 1990s, large numbers of minors characterised by the absence of a legal guardian have been arriving in these cities, in addition to the habitual immigrants already arriving [6,7]. Given the high number of minors making up the migratory phenomenon seen in Spain and the rest of the European continent, the European Council [8] classified these children for the first time as unaccompanied foreign minors (UFMs) and introduced an action protocol for host countries. The purpose of this is to guarantee the fundamental rights of UFMs. Institutional care in reception centres has been presented as the most effective means of offering a life free of violence and guaranteeing the social integration of these minors [9].

UFM often come from very large families living in extreme poverty. They arrive at destination countries without any legal guardian, often due to loss, or because they are escaping a situation with extenuating circumstances with the family’s consent to travel alone [6,7,10].

This situation makes immigrants exceptionally vulnerable. The scarcity of resources, low probability of success, behavioural risks, poor lifestyles and difficulties in accessing health services both conditions and determines the vulnerability of immigrants [2,11,12]. In addition to this, these children also have further challenges posed by their own cultural distancing, language difficulties, bad previous educational experiences, socialisation difficulties, lack of adaptation and problematic behaviours, alongside the physical and psychological effects of poor health or trauma [13,14]. These minors present high expectations in relation to their new life once they arrive in Spain. Failing to meet these expectations leads them to generate feelings of disappointment, frustration, rage and inferiority complexes [15].

With respect to this, and despite the fact that institutional attendance of these minors is positively reflected by the European Council [8], many UFMs live undocumented, without having been recorded in any migratory register [16,17,18]. This fact increases further still the vulnerability of these minors, which, according to some research studies [17,19], conditions and exposes the immigrant population to violent and aggressive situations. Together with other factors, this generates a dangerous and unsafe situation for the recipient population [5,6,12,13,14,20,21]. This is translated into a negative perception of UFM, accompanied by feelings of xenophobia and discrimination [1,22,23,24]. It also explains the feelings of rejection and social exclusion felt by these children [25,26,27].

Given this situation, a number of research studies have centred on better understanding the vital development and the political processes underlying the social integration of these minors [28,29,30,31]. However, studies linking educational aspects to the integration phenomenon of these children are scarce [32,33,34] and do not specify the educational tools and strategies that are most influential in regards to the social inclusion of UFM. Research conducted by Martínez-Martínez et al. [35] stands alone in demonstrating the positive influence of using certain strategies and tools on the personal learning environments (PLE) created by these minors for their educational and social integration. More specifically, the authors concluded that through the use of determined PLE’s, UFM improved their self-concept, planning and management of learning, use of educational strategies and tools, and social relationships within the educational setting. All of these are fundamental aspects of the social and academic integration of these students [36] and of the students who will form the basis of the present research.

PLE’s relate to the process through which students administer the information, resources and relationships they come across when engaging in learning [37,38]. In this way, PLE’s will promote the learning of students [39,40]. Further, technology use in the present day means that PLE’s should also be considered as platforms incorporating different tools, or management and communication applications [41,42]. This leads students to assume a digital identity when considering meaningful questions in relation to asynchronistic self-directed learning [43].

In considering conceptualisation of the terminology, many research studies exist which have related determinants of the PLE with improvements to the learning self-concept of ethnic minority students [44,45,46]. Similarly, studies have considered collaborative learning, timeless access to resources and interactions within the educational setting, which impact students’ self-concept. This aspect is especially relevant when we consider that self-concept is tightly related to the wellbeing of minors [47,48]. Indeed, improvements in self-concept will reduce vulnerability and feelings of frustration and rage in immigrants following arrival at their welcome country [13,14]. Further, this improves students’ attitudes towards effort and educational expectations, which will, in turn, influence their academic performance [49].

In addition, considering that adolescence is a complicated stage for self-concept [50,51], a number of research studies consider age as a determining factor of self-concept [52,53]. It should also be kept in mind that as these students grow, they will acquire greater knowledge about the metacognitive learning processes behind their learning [54,55].

When we speak about the planning and management of learning, terms emerge that are intrinsic to the concept of PLE’s when we consider that these promote self-directed learning [56]. With respect to this, research studies exist [57] that consider PLE’s as situational drivers for carrying out the planning of learning. It is especially important to consider this aspect in UFM, where the social links of planning the learning process are considered [58,59], alongside the social exclusion that these minors may end up experiencing [27]. In this way, Davoudi [60] indicates that during the planning of learning, students will consider what aspects they learn and from whom. All of this occurs through relationships formed with the educational setting.

Further, as was the case with the dimension of self-concept, studies exist that have determined age as an influential factor in the process of planning and managing learning [61,62].

In the baseline research study [35], PLE use was also related to access to educational strategies and tools. In this sense, multimedia learning resources currently take on special importance. These resources, often used through mobile technology [63,64], have been demonstrated to be effective in providing opportunities for communication amongst peers. This decreases social exclusion among minority students [65].

These types of tools and strategies provide timeless learning situations without the need for a fixed physical space [66,67]. They complement the traditional resources typically used by students [68].

Such resources are more easily utilised with secondary school students aged between 12 and 16 years. This is not only because, at this age, students are more skilled [69], but because they have greater access and spend more time using these technologies than younger students [70,71].

Finally, the baseline study also related PLE use with the social relationships engaged in within the student’s environment. This identified the social dimension of PLE’s, indicating that the use of tools and strategies in the learning process facilitates the creation of knowledge construction spaces. This is seen to take place through interaction with the setting in which learning communities are denominated [72,73]. Continuing with multimedia resources, the use of social networks such as, for example, Facebook, Instagram, WhatsApp and Snapchat, consequently, facilitate the creation of these communities [74,75]. Within these communities, in addition to sharing educational content, students can express feelings [76] and interact autonomously in communication processes [77].

In addition, the creation of these learning communities guarantees community cohesion [73,78]. This will help students to participate in communication with peers, little by little, until they achieve full integration in the learning communication [79]. This is especially relevant for UFM, given the socialisation challenges they face [32]. Further, in the same way, as occurs with the aforementioned dimensions, age is considered to be a determining factor when establishing social relationships. This is because social maturation increases with age and is a crucial aspect when establishing and maintaining friendships [80].

In consideration of the trajectory followed by previously conducted research studies on this topic, the present study aims to examine whether age-related psychosocial factors and the use of applications and social networks are determinants of the personal learning environments of unaccompanied foreign minors. The analysis was conducted taking four factors and the dimensions they compose into account: self-concept of the learning process, planning and management of learning, use of resources and tools, and communication and social interaction.

## 2. Materials and Methods

### 2.1. Design and Participants

The present research counted on a non-experimental design, which was descriptive and cross-sectional in nature. The population consisted of unaccompanied foreign minors (UFM) residing in the cross-border cities of Ceuta and Melilla. According to the Ministry of the Interior, whilst this deals with a highly variable population in number, the number of UFM attending reception centres in 2018 (period of data collection) ranged from 407 in Ceuta to 1090 in Melilla.

Considering the particularities of this population, we opted to invite the entire population group to participate. Cases were nonetheless selected in a random way, not corresponding to convenience sampling.

The sample is composed of 624 individuals (♂ = 92.1% (*n* = 575); ♀ = 7.9% (*n* = 49)) aged between 8 and 17 years (8–10 = 5.1% (*n* = 32); 11–13 = 11.4% (*n* = 71); 14–16 = 36.9% (*n* = 230); 17 = 46.6% (*n* = 291)), with the majority coming from Morocco (85%) and residing in the cities of Melilla (71.9%) and Ceuta (29.1%). When considering the formula established by Tagliacarne [81] for a finite population (1968), the sample is representative of the population group (*n* = 1497).

### 2.2. Instruments

The “PLE and Social Integration of UFM” questionnaire [35] was employed for data collection. This is composed of 39 rating scale items, with 6 response options grouped according to 4 factors: (1) self-concept of the learning process (SLP; 15 items), (2) planning and management of learning (PML; 10 items), (3) use of resources and tools (URT; 8 items), and (4) communication and social interaction (CSI; 6 items). It presents strong reliability with a Cronbach alpha coefficient of 0.897.

### 2.3. Procedure

In the first instance, necessary permissions were requested from the relevant administrations to be able to access the youth centres at which participating minors were registered (one in Ceuta and three in Melilla). To proceed with this phase, a positive outcome was also received from the Ethics Committee of the University of Granada (reference code: 742/CEIH/2018). With regards to the administration of the questionnaire, this was provided in paper format and in Spanish. Minors worked in small groups to complete the questionnaire and received the help of a translator. The whole process conformed to the ethical principles for research defined in the Declaration of Helsinki in 1975 and later updated in Brazil in 2013.

### 2.4. Data Analysis

Collected data were quantitative in nature and so analysis also employed a quantitative procedure. Concretely, frequencies and percentages were used to provide basic descriptive data, whilst one-factor ANOVA (for the “age” variable) and Student’s *t*-test for independent samples (for the “applications use” and “use of social networks” variables) were employed to examine relationships between variables. Instrument reliability was studied through Cronbach’s alpha coefficient. All analyses were conducted using the statistical program IBM SPSS in its version 26.0 (IBM Company, Armonk, NY, USA). Normality of the data was carried out with the Kolmogorov–Smirnov test. Levene’s test was used to Know homoscedasticity. The reliability index was established at 95.5%. The significance level was set at 0.05 [82].

## 3. Results

Firstly, results produced from the ANOVA test for the “age” variable will be presented. This analysis enabled us to uncover whether significant differences exist between the various age groups in relation to the four dimensions comprising minors’ PLEs.

Table 1 shows the first dimension: “self-concept of the learning process” (SLP). Statistically significant differences were found (0.000), with the highest values being obtained for UFM aged between 14 and 16 years, followed by those aged 17 years, those aged 11–13 years, and, finally, those aged 8–10 years (4.03 ± 0.64 vs. 4.00 ± 0.58 vs. 3.66 ± 0.62 vs. 2.96 ± 1.05; *p* < 0.001).

In Table 2, between-group (Bonferroni) differences can be observed for this dimension. Concretely, statistically significant differences exist between the following age groups: 8–10 and 11–13; 8–10 and 14–16; 8–10 and 17; 11–13 and 14–16; 11–13 and 17. In this sense, a comparison of the 14–16- and 17-year-old age groups was the only one to not produce a statistically significant outcome.

Table 1 shows the “planning and management of learning” (PML) dimension. Statistically significant differences were found (0.000), with the highest value being obtained for UFM aged between 14 and 16 years, followed by those aged 17 years, those aged 11–13 years, and, finally, those aged 8–10 years (3.90 ± 0.79 vs. 3.64 ± 0.99 vs. 3.48 ± 0.78 vs. 2.80 ± 0.94; *p* < 0.001).

For the PML dimension, statistically significant between-group differences (Bonferroni) were found for the following age groups: 8–10 and 11–13; 8–10 and 14–16; 8–10 and 17; 11–13 and 14–16; 14–16 and 17. In contrast, statistically significant differences were not found between the 11–13 age group and those aged 17 years (Table 2).

Statistically significant differences were found (0.000) for the third dimension, “use of resources and tools” (URT) (Table 1), with the highest value being obtained for UFM aged between 14 and 16 years, followed by those aged 17 years, those aged 11–13 years, and, finally, those aged 8–10 years (3.35 ± 0.83 vs. 3.32 ± 0.81 vs. 3.16 ± 0.66 vs. 2.65 ± 0.85; *p* < 0.001).

In Table 2, between-group differences (Bonferroni) can be observed for this dimension. Concretely, statistically significant differences exist between the following age groups: 8–10 and 11–13; 8–10 and 14–16; 8–10 and 17. In contrast, statistically significant differences were not found between the 11–13 and 14–16 age groups; 11–13 and 17 age groups; and 14–16 and 17 age groups.

Table 1 shows the “communication and social interaction” (CSI) dimension. Statistically significant differences were found (0.000), with the highest value being obtained for UFM aged between 14 and 16 years, followed by those aged 17 years, those aged 11–13 years, and, finally, those aged 8–10 years (3.78 ± 0.77 vs. 3.69 ± 0.80 vs. 3.52 ± 0.80 vs. 3.04 ± 0.84; *p* < 0.001).

For the CSI dimension (Table 2), statistically significant differences were found for the following age groups (Bonferroni): 8–10 and 14–16; 8–10 and 17. In contrast, statistically significant differences were not found between the following groups: 8–10 and 11–13; 11–13 and 14–16; 11–13 and 17; 14–16 and 17.

Secondly, the results of Student’s *t*-test (independent samples) are presented for the “use of apps” and “use of social networks” variables. These analyses permit us to better understand whether significant differences exist between those who use these resources and those who do not, in relation to the four dimensions that compose minors’ PLEs.

As can be observed in Table 3, statistically significant differences were uncovered for three of the four dimensions composing the PLE. Specifically, UFMs who use apps reported higher scores for the planning and management of learning (PML; 3.85 ± 0.76 vs. 3.23 ± 1.15, *p* < 0.001), the use of resources and tools (URT; 3.40 ± 0.77 vs. 2.97 ± 0.88, *p* < 0.001), and for communication and social interaction (CSI; 3.75 ± 0.75 vs. 3.50 ± 0.91, *p* = 0.004).

Table 3 shows the associations between the use of social networks and PLE dimensions in UFMs. Statistically significant differences were found in relation to the four dimensions and, in all cases, minors who made use of social networks scored better: self-concept of the learning process (SLP; 4.03 ± 0.60 vs. 3.84 ± 0.73, *p* = 0.002), planning and management of learning (PML; 3.94 ± 0.72 vs. 3.46 ± 1.01, *p* < 0.001), use of resources and tools (URT; 3.52 ± 0.74 vs. 3.08 ± 0.83, *p* < 0.001), and communication and social interaction (CSI; 3.89 ± 0.71 vs. 3.50 ± 0.84, *p* < 0.001).

## 4. Discussion

In relation to the self-concept held by UFMs in relation to their learning processes as a function of age, statistically significant differences were found in relation to older age groups. Highest scores were reported by those aged between 14 and 16 years old, followed by those aged 17 years, with very little variation being observed in mean scores. In this way, other authors produced similar data [52]. In corroboration with the present study, their study found that 12–15-year-old adolescents were quicker to understand, completed their work to a higher standard and were more capable of rationalising what they learned. Likewise, other research [53] adds that younger individuals are more highly motivated and report intrinsic motivation to be their more dominant motivational type.

On the other hand, more complex skills are not acquired until the age of 18. Such skills include expressing what one is good at and where they have the most difficulty acting [52]. This is due to the fact that older individuals have greater understanding and awareness of knowledge about themselves, greater capacity for meta-cognitive reflection, better motivation–action–results for their way of learning, and better academic performance. In addition, they take more action, engage in better decision making and put more effective strategies in place in response to outcomes of the teaching-learning process [52,54,55,84,85,86,87].

Nevertheless, it must be kept in mind that these ages correspond to a critical life stage. Numerous authors indicate that academic problems can appear at this stage and there is a possibility that adolescents will abandon their studies due to a lack of interest, lack of goals in their life project or influences exerted by their peer group [61,62,88,89,90]. This situation is seen to be accentuated in UFM as uncertainty surrounding their academic and professional future directly depends on their permanent and administrative situation in their recipient country. In this sense, it is necessary to act during this life-stage due to the influence it has on academic management. For this reason, it is crucial to increase intrinsic motivation levels in adolescents as this promotes positive functional self-concept.

With regards to the use of resources and tools, results produced differences in relation to age, with minors in the 14–16-year age group scoring most highly. Other research studies [80,91,92,93] reflect similar results, revealing that information use, in a self-organised way, has become a necessary skill for controlling learning and academic performance in an effective way. This may be due to the fact that, at this age, personal learning processes (PLEs) are more effective with regards to the virtual settings to which they relate. As a consequence, adolescents show a greater mastery of technology and, thus, are more able to apply it appropriately.

Along these lines, results of the present research study show that use of applications and social networks is significant in the construction of minors’ personal environments. Recent research confirms these results, suggesting that virtual learning environments are understood as spaces which favour learning [94]. Even other research [95] provides evidence that smartphone use as a classroom tool improves pedagogical reach and helps users to become more familiarised with devices. In the same way, a recent research [96] reports a high degree of acceptance of communication via WhatsApp, by students who stated it was useful for clarifying doubts, sharing documents and information, and interacting with classmates and peers. This is seen in such a way that the inclusion of UFMs on the right side of this digital divide is viable and would promote their motivation towards teaching-learning processes.

Finally, the capacity for communication and social interaction also depends significantly on the minor’s age. In the same way as seen with the aforementioned dimensions, older age groups are also seen to score better on this dimension. This may be explained by the fact that older adolescents have developed greater social maturity, whilst also having a greater ability to make and keep friends [80]. Nonetheless, authors indicate that, despite enjoying a wide skillset, unaccompanied foreign minors who have spent more time residing in Spain, experience detriments to their communication and social processes with native individuals who possess similar characteristics. This complicates their educational inclusion, largely due to the strong rejection they generally suffer at the hands of social agents in their context [32,97,98,99].

Unaccompanied foreign minors arriving to Spain are mainly aged within the age range of 14–17 years old. This places them in the ideal moment to develop their intrinsic motivation and self-concept, whilst also reinforcing their use of both virtual and physical tools in order to improve their learning processes. In this way, they will be more prepared to develop self-regulated knowledge, which will guide them through academic timetables and in their transition towards active life.

## 5. Conclusions

In conclusion, of the main outcomes extracted from the present research, it must be highlighted that for the four dimensions inherent to the questionnaire, significant differences were found as a function of age. The same trend was observed within each dimension, with older age groups scoring more highly. In all cases the 14–16 age group scored best, followed by those age 17 years, without relevant mean differences existing between the scores of these groups. In fact, when examining between-group differences (Bonferroni) for the different dimensions, it is observed that no significant differences exist between these two age groups with regards to self-concept of the learning process, use of learning tools and resources, and the capacity to communicate and socially interact. Further, results show that the use of applications and social networks is significant and favours the construction of minors’ personal environments.

It is also important to outline the limitations of the present study. Difficulties accessing the sample must be highlighted. Nevertheless, reliable data were collected, counting on the collaboration of various institutions and the participation of a high number of researchers. In fact, the results are transferable to the general national population. Results may be used with the aim of improving the personal learning environments of these minors, whilst aiding their integration in the educational and social setting.

## Figures and Tables

**Table 1 ijerph-17-03700-t001:** Relationships between personal learning environments (PLE) factors and the age of unaccompanied foreign minors (UFMs).

PLE Factors	Age Group	M	SD	CI (95%)		F	Sig.
				Lower Limit	Higher Limit		
SLP	8–10 years	2.96	1.05	2.52	3.41	23.517	0.000 *
	11–13 years	3.66	0.62	3.49	3.84		
	14–16 years	4.03	0.64	3.93	4.12		
	17 years	4.00	0.58	3.92	4.07		
PML	8–10 years	2.80	0.94	2.42	3.18	13.640	0.000 *
	11–13 years	3.48	0.78	3.28	3.68		
	14–16 years	3.90	0.79	3.79	4.01		
	17 years	3.64	0.99	3.51	3.76		
URT	8–10 years	2.65	0.85	2.32	2.98	6.902	0.000 *
	11–13 years	3.16	0.66	2.99	3.33		
	14–16 years	3.35	0.83	3.24	3.46		
	17 years	3.32	0.81	3.22	3.42		
CSI	8–10 years	3.04	0.84	2.69	3.39	7.424	0.000 *
	11–13 years	3.52	0.80	3.31	3.73		
	14–16 years	3.78	0.77	3.68	3.89		
	17 years	3.69	0.80	3.59	3.79		

Note: * Difference is significant at the level of 0.05.

**Table 2 ijerph-17-03700-t002:** Significant between-group differences for PLE factors (Bonferroni).

PLE Factors	(I) Age	(J) Age	Mean Difference (I–J)	Stand. Error	Sig.	ES (d)	CI (95%)	
							Lower Limit	Upper Limit
SLP	8–10	11–13	−0.70 *	0.16	0.000	0.900	0.465	1.335
		14–16	−1.06 *	0.14	0.000	1.525	1.133	1.917
		17	−1.03 *	0.14	0.000	1.623	1.238	2.009
	11–13	14–16	−0.36 *	0.10	0.002	0.582	0.312	0.852
		17	−0.33 *	0.10	0.005	0.578	0.315	0.841
	14–16	17	0.03	0.06	1.000	0.049	−0.124	0.222
PML	8–10	11–13	−0.68 *	0.21	0.007	0.817	0.385	1.249
		14–16	−1.10 *	0.19	0.000	0.832	0.971	1.747
		17	−0.84 *	0.18	0.000	0.853	0.482	1.223
	11–13	14–16	−0.42 *	0.13	0.010	0.533	0.264	0.803
		17	−0.16	0.13	1.000	0.168	−0.092	0.428
	14–16	17	0.26 *	0.09	0.014	0.287	0.113	0.46
URT	8–10	11–13	−0.50 *	0.18	0.039	0.705	0.276	1.133
		14–16	−0.70 *	0.16	0.000	0.841	0.464	1.218
		17	−0.67 *	0.16	0.000	0.823	0.453	1.194
	11–13	14–16	−0.19	0.12	0.594	0.239	−0.027	0.506
		17	−0.17	0.12	0.875	0.204	−0.056	0.464
	14–16	17	0.03	0.08	1.000	0.037	−0.136	0.21
CSI	8–10	11–13	−0.48	0.19	0.069	0.591	0.166	1.016
		14–16	−0.74 *	0.17	0.000	0.950	0.572	1.329
		17	−0.65 *	0.17	0.001	0.809	0.438	1.179
	11–13	14–16	−0.26	0.12	0.155	0.335	0.067	0.602
		17	−0.17	0.12	0.798	0.212	−0.047	0.472
	14–16	17	0.09	0.07	1.000	0.114	−0.059	0.287

Note 1: * Mean difference is significant at the level of 0.05. Note 2: The magnitude of the differences (effect size; ES) was obtained using Cohen’s standardized measure d [83], interpreted as null (0–0.19), low (0.20–0.49), moderate (0.50–0.79), or high (≥0.80). Thus, the 95% confidence interval (95% CI) was calculated for each effect size.

**Table 3 ijerph-17-03700-t003:** Relationships between the use of apps and the use of social networks and PLE dimensions.

			M	SD	Standard Error	F	Sig. (Levene)	Sig. (Two-tailed)	ES (d)	CI (95%)	
										Lower Limit	Upper Limit
**Use of Apps**	SLP	Yes	3.95	0.66	0.035	1.741	0.188	0.132	0.065	−0.133	0.263
		No	3.85	0.72	0.062						
	PML	Yes	3.85	0.76	0.039	56.286	0.000 *	0.000*	0.700	0.503	0.897
		No	3.23	1.15	0.096						
	URT	Yes	3.40	0.77	0.038	5.996	0.015 *	0.000*	0.505	0.315	0.695
		No	2.97	0.88	0.072						
	CSI	Yes	3.75	0.75	0.038	4.011	0.046 *	0.004 *	0.314	0.125	0.502
		No	3.50	0.91	0.074						
**Use of social networks**	SLP	Yes	4.03	0.60	0.040	9.770	0.002 *	0.002 *	0.282	0.105	0.458
		No	3.84	0.73	0.044						
	PML	Yes	3.94	0.72	0.047	24.182	0.000 *	0.000*	0.539	0.363	0.715
		No	3.46	1.01	0.060						
	URT	Yes	3.52	0.74	0.047	4.922	0.027 *	0.000 *	0.557	0.385	0.728
		No	3.08	0.83	0.048						
	CSI	Yes	3.89	0.71	0.045	3.982	0.047 *	0.000 *	0.498	0.327	0.669
		No	3.50	0.84	0.049						

Note 1: * Difference is significant at the level of 0.05. Note 2: The magnitude of the differences (effect size; ES) was obtained using Cohen’s standardized measure d [83], interpreted as null (0–0.19), low (0.20–0.49), moderate (0.50–0.79), or high (≥0.80). Thus, the 95% confidence interval (95% CI) was calculated for each effect size.

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
