# Peer review of "An Analysis of Personal Learning Environments and Age-Related Psychosocial Factors of Unaccompanied Foreign Minors"

_ijerph, 2020, doi:10.3390/ijerph17103700_

Round 1

Reviewer 1 Report

Dear authors,

First, I would like to thanks the editor to send me this manuscript for peer review. I would like also to thank the authors for their massive work in this investigation.

The introduction section tries to summarize the existing state of the art. The introduction is concise and enough. The most relevant previous studies are cited and the importance of this investigation is pointed out across this section.

The methods section contains all the information necessary to understand the procedures of the investigation and to ensure its reproducibility. The sample size should be highlighted given the population of this study.

The results section responds to the objective of this research. The tables show the information most relevant to answer the research question.

In sum, the manuscript is well written and the conclusion is supported for the results. The design of the study carried out according to scientific standards. The sample size is very high according to the target population. The results are meaningful given the novelty of the investigation. The quality of the manuscript is aligned with the quality standards of this journal. As a minor revision, I would encourage authors to add tables’ notes to better understand the results.

Sincerely,

Author Response

Dear reviewer,

We would like to express our gratitude for the time taken to review this manuscript and for the comments made, which we believe to be critical for producing rigorous and quality research. We have detailed below the changes made in the original article: “An analysis of personal learning environments and age-related psychosocial factors of unaccompanied foreign minors” (ijerph-).

Modifications have been made in the original manuscript following the reviewers’ comments. For each modification we have written: the original comment as written by the reviewer; and the change made in response to that comment. Changes have been made using the tool “Track changes” enabling editor and reviewers to identify modifications easily.

Comment 1:

As a minor revision, I would encourage authors to add tables’ notes to better understand the results.

Response 1:

Notes have been included in the tables to better understand the results. Review manuscript pages 6 and 7.

Reviewer 2 Report

Overall, the proposal is very informative, adds value to the field. Processing data was done well, and the results are well discussed. There are a few notes which I wish to raise, hoping that they are useful to authors to improve their work.

The abbreviations used in the whole manuscript is very confusing. I invite authors to limit using abbreviations in the abstract. What is "MENA" abbreviating? MENA and UFM are used as if they are synonyms. Please choose only one between MENA and UFM and use it consistently in the whole manuscript.

Another note is that all presented tables seem to be copy-pasted from the output of SPSS without much wisdom. This makes the results be presented in a very confusing way. For example, in table 2, the negative sign for "mean difference" is meaningless. You have four age groups: 8-10, 11-13, 14-16, and 17. Then when you process the Tukey-Kramer multiple comparison procedure with your ANOVA results, you only have to compare groups as below:

1) Comparing "8-10" with "11-13", "14-16", and "17"

2) Comparing "11-13" with "14-16" and "17"

3) Comparing "14-16" with "17"

Please note that, for example, comparing "8-10" with "11-13" gives the same results with the comparison of "11-13" with "8-10."

I invite authors to present their results in a more innovative way rather than just showing them as they appear in the output window of SPSS.

Author Response

Dear reviewer,

We would like to express our gratitude for the time taken to review this manuscript and for the comments made, which we believe to be critical for producing rigorous and quality research. We have detailed below the changes made in the original article: “An analysis of personal learning environments and age-related psychosocial factors of unaccompanied foreign minors” (ijerph-).

Modifications have been made in the original manuscript following the reviewers’ comments. For each modification we have written: the original comment as written by the reviewer; and the change made in response to that comment. Changes have been made using the tool “Track changes” enabling editor and reviewers to identify modifications easily.

Comment 1.

The abbreviations used in the whole manuscript is very confusing. I invite authors to limit using abbreviations in the abstract. What is "MENA" abbreviating? MENA and UFM are used as if they are synonyms. Please choose only one between MENA and UFM and use it consistently in the whole manuscript.

Response 1:

The use of abbreviations in the abstract has been limited. Only the abbreviations that appear in the name of the instrument have been kept, since we cannot modify it. Similarly, the abbreviation MENA has been removed throughout the text and has been replaced by UFM. In this way the discourse has been homogenized.

Comment 2:

Another note is that all presented tables seem to be copy-pasted from the output of SPSS without much wisdom. This makes the results be presented in a very confusing way. For example, in table 2, the negative sign for "mean difference" is meaningless. You have four age groups: 8-10, 11-13, 14-16, and 17. Then when you process the Tukey-Kramer multiple comparison procedure with your ANOVA results, you only have to compare groups as below:

1) Comparing "8-10" with "11-13", "14-16", and "17"

2) Comparing "11-13" with "14-16" and "17"

3) Comparing "14-16" with "17"

Please note that, for example, comparing "8-10" with "11-13" gives the same results with the comparison of "11-13" with "8-10."

I invite authors to present their results in a more innovative way rather than just showing them as they appear in the output window of SPSS.

Response 2:

The tables have not been directly copied from the SPSS program, on the contrary the tables have been prepared in word. For its preparation, an article published in this same editorial (MDPI) was taken as reference:

Chacón-Cuberos, R., Zurita-Ortega, F., Martínez-Martínez, A., Olmedo-Moreno, E.M, & Castro-Sánchez, M. (2018). Adherence to the Mediterranean Diet Is Related to Healthy Habits, Learning Processes, and Academic Achievement in Adolescents: A Cross-Sectional Study. Nutrients, 10, 1-13. doi:10.3390/nu10111566

However, following the indications of the reviewer, the tables have been modified and new elements have been incorporated. Finally, the results have been merged into three tables (pages 5-7).

And as the reviewer indicates, for the table in which the Bonferroni results are included, we have only compared groups as below (Table 2, page 6):

1) Comparing "8-10" with "11-13", "14-16", and "17"

2) Comparing "11-13" with "14-16" and "17"

3) Comparing "14-16" with "17"

As the data collected in the research have a normal distribution and meet the criteria for the ANOVA test, it has been considered appropriate to use Bonferroni to make the comparison between groups.

The following text fragment has been included in “data analysis” section to clarify (page 4):

“Normality of the data was carried out with the Kolmogorov-Smirnov test. Levene’s test was used to Know homoscedasticity. The reliability index was established at 95.5%. The significance level was set at 0.05.” [82]

Reviewer 3 Report

The statistical treatment is weak, and it would be necessary to show some more data.

I strongly recommend presenting the results of the Levene statistic as evidence of homogeneity, since the standard deviation results are very low, which indicates very little dispersion in the data.   It would be important to present the results of the magnitude of the effect, since the p-value represents only if the value is "statistically" significant, but does not indicate the size.   I also recommend presenting only one table with the results of the ANOVAS and another with the Student's t-test including the four factors along with the results of the Levene statistics.  The results of the multiple comparisons (Bonferroni) can be left separated by the factors, but should include the effect sizes.

Author Response

Dear reviewer,

We would like to express our gratitude for the time taken to review this manuscript and for the comments made, which we believe to be critical for producing rigorous and quality research. We have detailed below the changes made in the original article: “An analysis of personal learning environments and age-related psychosocial factors of unaccompanied foreign minors” (ijerph-).

Modifications have been made in the original manuscript following the reviewers’ comments. For each modification we have written: the original comment as written by the reviewer; and the change made in response to that comment. Changes have been made using the tool “Track changes” enabling editor and reviewers to identify modifications easily.

Comment 1.

I strongly recommend presenting the results of the Levene statistic as evidence of homogeneity, since the standard deviation results are very low, which indicates very little dispersion in the data. It would be important to present the results of the magnitude of the effect, since the p-value represents only if the value is "statistically" significant, but does not indicate the size.  I also recommend presenting only one table with the results of the ANOVAS and another with the Student's t-test including the four factors along with the results of the Levene statistics.  The results of the multiple comparisons (Bonferroni) can be left separated by the factors, but should include the effect sizes.

Response 1:

Following the recommendations of reviewer:

- Three new tables have been generated that unify the results of the ANOVA (table 1), the results of the multiple comparisons -Bonferroni (table 2) and the results of Student's t-test (table 3) (pages 5-7).

- The results of the Levene statistic as evidence of homogeneity have been included in the table 3 (Student's t-test) (page 7).

- The results of the multiple comparisons (Bonferroni) include the effect sizes (table 2) (page 6).

- The effect sizes have been included in the table 3 (Student's t-test) (page 7).